# Engineered proteins with sensing and activating modules for automated reprogramming of cellular functions

Jie Sun[1,2,3], Lei Lei[4], Chih-Ming Tsai[5], Yi Wang[2], Yiwen Shi[4], Mingxing Ouyang[4], Shaoying Lu[4], Jihye Seong[6], Tae-Jin Kim[6,7], Pengzhi Wang[4], Min Huang[2], Xiangdong Xu[8], Victor Nizet [5], Shu Chien[4] & Yingxiao Wang[1,2,4,6]

Protein-based biosensors or activators have been engineered to visualize molecular signals or manipulate cellular functions. Here we integrate these two functionalities into one protein molecule, an integrated sensing and activating protein (iSNAP). A prototype that can detect tyrosine phosphorylation and immediately activate auto-inhibited Shp2 phosphatase, Shp2-iSNAP, is designed through modular assembly. When Shp2-iSNAP is fused to the SIRPα receptor which typically transduces anti-phagocytic signals from the 'don't eat me' CD47 ligand through negative Shp1 signaling, the engineered macrophages not only allow visualization of SIRPα phosphorylation upon CD47 engagement but also rewire the CD47-SIRPα axis into the positive Shp2 signaling, which enhances phagocytosis of opsonized tumor cells. A second SIRPα Syk-iSNAP with redesigned sensor and activator modules can likewise rewire the CD47-SIRPα axis to the pro-phagocytic Syk kinase activation. Thus, our approach can be extended to execute a broad range of sensing and automated reprogramming actions for directed therapeutics.

[1] Beckman Institute for Advanced Science and Technology, University of Illinois at Urbana-Champaign, Urbana, IL 61801, USA. [2] Department of Bioengineering, University of Illinois at Urbana-Champaign, Urbana, IL 61801, USA. [3] Department of Cell Biology, School of Medicine, Zhejiang University, Hangzhou 310058, China. [4] Department of Bioengineering and Institute of Engineering in Medicine, University of California, San Diego, La Jolla, CA 92093, USA. [5] Department of Pediatrics and Skaggs School of Pharmacy and Pharmaceutical Sciences, University of California, San Diego, La Jolla, CA 92093, USA. [6] Neuroscience Program, University of Illinois at Urbana-Champaign, Urbana, IL 61801, USA. [7] Department of Biological Sciences, Pusan National University, Busan 46241, Republic of Korea. [8] Department of Pathology, Veterans Affairs San Diego Healthcare System, University of California, San Diego, La Jolla, CA 92093, USA. Jie Sun and Lei Lei contributed equally to this work. Correspondence and requests for materials should be addressed to S.C. (email: shuchien@ucsd.edu) or to Y.W. (email: yiw015@eng.ucsd.edu)

Numerous genetically encoded biosensors based on fluorescent proteins (FPs) and fluorescence resonance energy transfer (FRET) have been generated to visualize dynamics of signal transduction in live cells[1]. In parallel, synthetic proteins that can undergo dimerization or allosteric conformational change upon stimulation by radio wave, light, chemical compounds or cell–cell interactions have been designed to activate signal cascades and control cellular behavior[2–5]. However, these two separate approaches have not been integrated into one platform to engineer proteins with both sensing and activating functions. We have developed a novel approach of using integrated sensing and activating proteins (iSNAPs) to surveil the intracellular space, and to immediately trigger designed molecular actions upon detection and visualization of specific signals, with the consequences of modulating the downstream signaling cascades and cellular functions.

Cancer immunotherapies targeting the patient's immune system, such as chimeric antigen receptor (CAR)-engineered T cells and immune checkpoint blockade represent a new direction of cancer treatment. Meanwhile, monoclonal antibody (mAb) therapies directly targeting cancer cells have been widely used[6]. Targeting both immune system and cancer, opsonin-dependent tumor cell phagocytosis mediated by macrophages provides an important effector mechanism for mAb-based cancer therapy[7]. Phagocytosis outcomes result from a tug-of-war between the pro-phagocytic (eat me) and anti-phagocytic (don't eat me) signals[8]. Antigen-targeting antibody can ligate FcγRs on macrophages and trigger pro-phagocytic signaling, while the anti-phagocytic signaling is mediated by CD47 on target cells engaging its receptor SIRPα on macrophages[9]. Therefore, anti-CD47 antibody has been applied to prevent anti-phagocytic signaling and promote tumor eradication in various cancer types[10–12]. CD47 is, however, expressed at high levels in red blood cells (RBCs), and undesirable anemia caused by CD47-blocking antibody-induced phagocytosis of RBCs may complicate anti-CD47 cancer therapy[13]. Hence, engineering macrophages with rewired CD47 signaling, together with monoclonal antibodies to specifically target tumor cells, can lead to a revolutionary development of next-generation immunotherapy.

Here we integrate protein modules with sensing and actuating functions to engineer iSNAPs capable of detecting tyrosine phosphorylation events and activating desired enzymatic functions. We apply these iSNAPs to rewire the 'don't eat me' CD47 signaling, leading to significantly enhanced phagocytic capabilities of engineered macrophages against tumor cells.

## Results
**Design and characterization of Shp2-iSNAP.** We have adopted a modular assembly approach to develop a general class of iSNAPs that can sense specific biochemical events and consequently activate the reprogramming of cellular functions (Supplementary Fig. 1a, b). We first engineered a Shp2-based iSNAP (Shp2-iSNAP) for the sensing of intracellular tyrosine phosphorylation and the consequent activation of a ubiquitous protein tyrosine phosphatase (PTP) Shp2[14]. Structurally, the enzymatic PTP domain of Shp2 is masked by the intramolecular auto-inhibitory N-SH2, which can be released upon binding of a phosphorylated peptide[14]. We have hence fused a phosphorylatable peptide, a FRET pair (ECFP as the donor and YPet as the acceptor), and Shp2 without its C-terminal tail to create Shp2-iSNAP. Shp2 C-tail contains two phosphorylatable tyrosines, which could interfere with the desired binding between N-SH2 and the phosphorylatable peptide, therefore, was eliminated (Fig. 1a). Upon kinase phosphorylation, the peptide can bind to the N-SH2 domain to cause FRET changes and

subsequently relieve the inhibitory effect of N-SH2 on the PTP domain, thus activating Shp2. Hence, FRET signals of the sensor module within iSNAPs can provide immediate readings and serve as reporters for functional calibration and optimization of engineered iSNAPs. Among the peptide sequences that we have tested (Supplementary Table 1), a BTAM peptide derived from the CD47 receptor SIRPα with two tyrosines led to an iSNAP that undergoes both FRET change (Fig. 1b, c) and PTP activation (Fig. 1d) in vitro upon phosphorylation by Src, a kinase known to phosphorylate SIRPα in response to various mitogens[15]. As designed, the mutation of these sensing tyrosines (FF mutation) in BTAM abolished phosphorylation, FRET response and PTP activation (Fig. 1c, d and Supplementary Fig. 2a). These results suggest that iSNAP can undergo intramolecular interactions and enzymatic activation precisely following our design.

We then tested the functionality of iSNAP in live mammalian cells. When Shp2-iSNAP was expressed in mouse embryonic fibroblasts (MEFs), platelet-derived growth factor (PDGF) caused a FRET increase (Fig. 1e, Supplementary Fig. 2b and Supplementary Movie 1), which depends on the phosphorylatable tyrosines (Fig. 1f). A clear increase of FRET (YFP channel) and a concomitant decrease of ECFP intensity of one representative MEF indicates FRET efficiency increase after PDGF addition (Fig. 1g). Therefore, our Shp2-iSNAP can sense and report PDGFR activation-induced tyrosine phosphorylation events in live cells. To further evaluate whether Shp2-iSNAP can rewire and activate designed downstream signaling events in MEFs, we utilized a FRET-based Lyn-FAK biosensor to monitor the membrane activity of FAK kinase (Fig. 1h), which can be dephosphorylated at Y397 by Shp2, leading to inhibited kinase activity[16]. To avoid interference between two pairs of ECFP and YPet in the same cell, we mutated the ECFP (T65A/W66A) and YPet (G66A) in the Shp2-iSNAP to eliminate its fluorescence while leaving other domains intact (Fig. 1h and Supplementary Fig. 3a). Shp2-iSNAP significantly reduced PDGF-induced FAK activity (Fig. 1i and Supplementary Fig. 3b). These results confirmed the designed functionality of Shp2-iSNAP in vitro and in live cells. This modular assembly approach was also adopted and extended to develop other phosphatase-based iSNAPs such as Shp1-iSNAP and Shp2/Shp1 hybrid iSNAP, which were confirmed to display desired FRET changes and PTP activation upon kinase stimulation (Supplementary Fig. 4).

**SIRPα Shp2-iSNAP rewires CD47-SIRPα axis in macrophages.** We further applied iSNAP for the regulation of cellular functions, in particular, macrophage phagocytosis. Shp2, opposite to Shp1, is widely considered to positively promote signaling related to receptor tyrosine kinases[17–19]. Specific for SIRPα, Shp2-SIRPα complex was shown to activate MAPK/ERK pathway in fibroblasts[15]. In macrophages, Shp2 was shown to activate Syk for pro-inflammatory response upon fungal infection[20] and positively regulate reactive oxygen species (ROS) through ERK activation to enhance phagocytic activities[21]. As such, Shp2 should play a positive role for macrophage phagocytosis. Because Shp2-iSNAP can detect the tyrosine phosphorylation of the BTAM motif from SIRPα, we fused Shp2-iSNAP at the C-tail of SIRPα (SIRPα Shp2-iSNAP) to rewire the anti-phagocytic 'don't eat me' CD47-SIRPα-Shp1-negative signaling[8, 17] into a positive Shp2 action[17, 21] upon the engagement of SIRPα by CD47, thus facilitating phagocytosis of opsonized tumor cells initiated by IgG-FcγR interactions (Fig. 2a). HEK293T cells gained the ability to bind to CD47 when expressing SIRPα Shp2-iSNAP, indicating its correct transmembrane localization and ligand-binding function (Supplementary Figs 5 and 6). Global inhibition of PTPs by pervanadate caused a drastic increase in the FRET signal of SIRPα Shp2-iSNAP but not

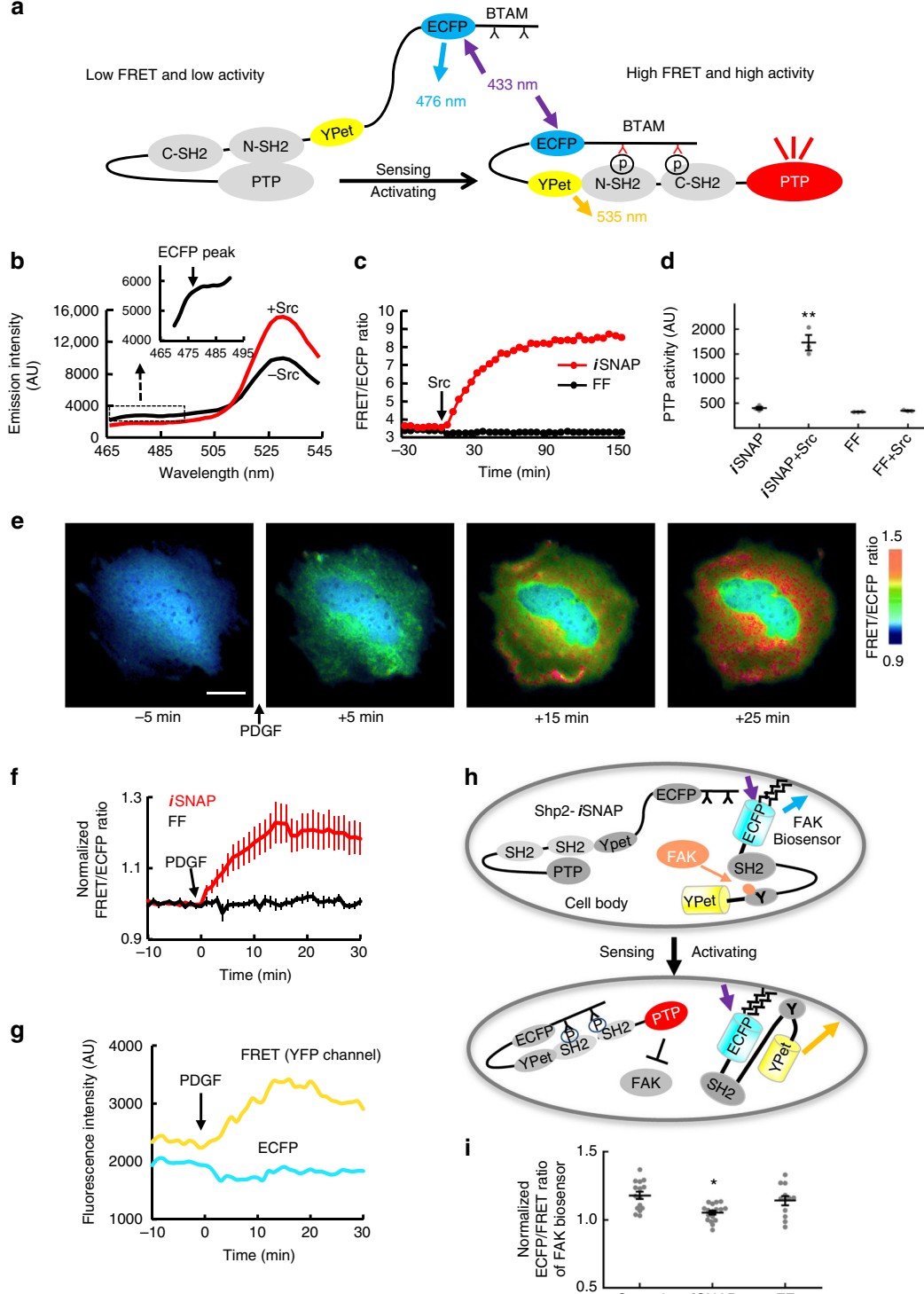

**Fig. 1** The design and characterization of the Shp2-*i*SNAP in vitro and in cells. **a** Schematic representation of the drawing of Shp2-*i*SNAP displaying a FRET increase and an activated PTP (*red*) after a conformational change induced by phosphorylation (P). **b** The emission spectrum change of the Shp2-*i*SNAP before (*black*) and after (*red*) Src kinase incubation in vitro. The inset shows the ECFP peak of the Shp2-*i*SNAP spectrum before kinase incubation. **c**, **d** The emission ratio time courses **c** and phosphatase activities **d** of Shp2-*i*SNAP and its FF mutant before and after Src incubation *in vitro* (n = 3, 3, 3, 3). **e** The ratiometric images of a MEF cell transfected with the Shp2-*i*SNAP before and after PDGF stimulation at indicated time points. All color *scale bars* in figures represent the FRET/ECFP emission ratio. *Scale bars*, 20 μm. **f** The ratio time courses (mean ± s.e.m.) of MEFs expressing the Shp2-*i*SNAP (n = 10) or its FF mutants (n = 7). **g** The time course of FRET and ECFP fluorescence intensities of one cell from **f**. A clear increase of FRET (YFP channel) and a concomitant decrease of ECFP intensity indicates FRET efficiency increase after PDGF addition. **h** Schematic drawing of the inhibitory effect of the activated non-fluorescent Shp2-*i*SNAP on FAK monitored by a membrane-bound FAK biosensor. The activated FAK and Shp2-*i*SNAP are colored in orange and red, respectively. **i** Dot plot of FAK biosensor emission ratios (mean ± s.e.m.) in PDGF-stimulated MEFs expressing control vector, the Shp2-*i*SNAP, or its FF mutant (n = 15, 17, 12). *P < 0.05. **P < 0.01 (two-sided Mann–Whitney U-test adjusted for multiple group comparison)

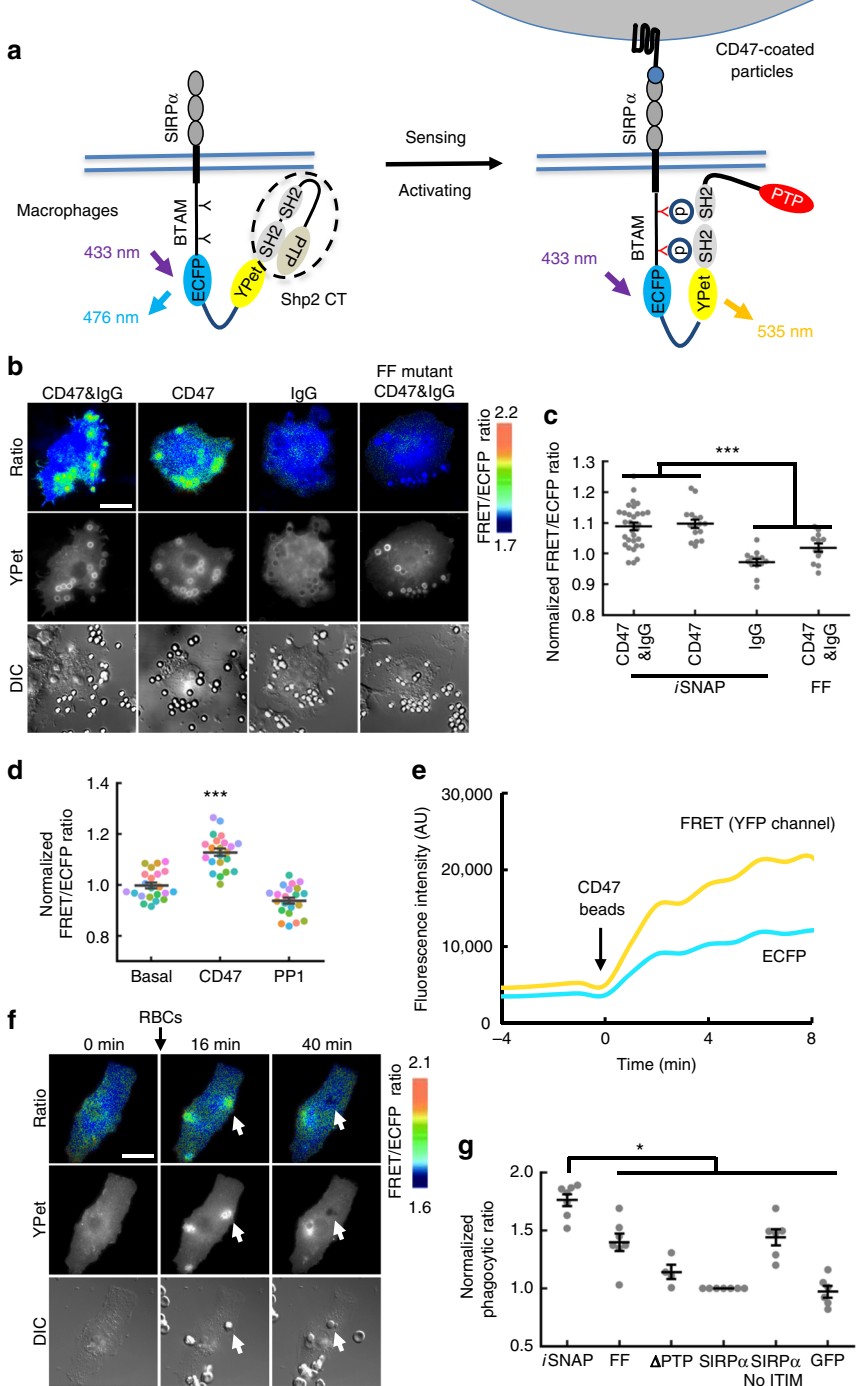

**Fig. 2** The activation and phagocytic functions of SIRPα Shp2-*i*SNAP in macrophages. **a** Schematic representation of the drawing of the SIRPα Shp2-*i*SNAP and its putative activation mechanism upon CD47 engagement. SIRPα Shp2-*i*SNAP contains a human SIRPα without its ITIM-containing C-tail, fused to the Shp2-*i*SNAP. **b** Images of RAW264.7 macrophages expressing SIRPα Shp2-*i*SNAP or FF mutant after incubation with beads coated by CD47 plus IgG, CD47 only, or IgG only, respectively. **c** Quantification of local FRET response in macrophages as represented in **b** ($n = 32, 16, 13, 12$). **d** The FRET ratio of SIRPα Shp2-*i*SNAP in response to CD47-coated beads and 10 μM SFKs inhibitor PP1 treatment in macrophages ($n = 22, 22, 22$). The same colored dots represent data obtained from the same cell. **e** The time course of FRET and ECFP fluorescence intensities of one cell from **d**. The increase of FRET intensity is greater than that of ECFP, so the calculated FRET/ECFP ratio increases compared to before CD47 beads addition. **f** Images of phagocytosis of opsonized RBCs by a representative RAW264.7 macrophage expressing SIRPα Shp2-*i*SNAP at indicated time points. Arrows indicate ingested RBCs. **g** Dot plot of normalized phagocytic rate (mean ± s.e.m.) of macrophages expressing SIRPα Shp2-*i*SNAP or its controls against opsonized RBCs ('FF', phenylalanine mutations replacing two tyrosines in the BTAM peptide; 'ΔPTP', SIRPα Shp2-*i*SNAP without PTP domain; 'SIRPα', full-length SIRPα fused with YPet; 'SIRPα-no ITIM', ITIM-truncated SIRPα fused with YPet) ($n = 7, 7, 4, 7, 6, 6$). *$P < 0.05$, ***$P < 0.001$ (two-sided Mann–Whitney *U*-test adjusted for multiple group comparison). *Scale bars*, 20 μm

its FF mutant, confirming the BTAM phosphorylation-dependent specificity as designed (Supplementary Fig. 7). We then examined Shp2-*i*SNAP in a macrophage cell line, RAW264.7. When CD47-coated beads engaged the RAW264.7 macrophages expressing SIRPα Shp2-*i*SNAP, a local FRET increase was observed (Fig. 2b, c and Supplementary Movie 2). This activation was specific to CD47 and depended on tyrosine phosphorylation, since it was abolished in groups using beads coated with IgG only, with FF mutant, or in the presence of PP1 to inhibit Src family-mediated tyrosine phosphorylation (Fig. 2b–d). CD47 beads induced SIRPα Shp2-*i*SNAP aggregation (Supplementary Fig. 21a), resulting an increase of both FRET and ECFP intensities (Fig. 2e), possibly due to a drastic change of structure and morphology at the proximal region of the beads, which led to the copy number change of the SIRPα Shp2-*i*SNAP at the local sites. The increase of FRET intensity, however, was obviously greater than that of ECFP, so the calculated FRET/ECFP ratio increased comparing to before CD47 beads addition, indicating an increase of FRET efficiency (Fig. 2e).

**SIRPα Shp2-*i*SNAP enhances phagocytosis of opsonized cells.**
Next, we examined RAW264.7 macrophage cell line engineered with SIRPα Shp2-*i*SNAP for their phagocytic activity against

target cells, e.g., human RBCs that endogenously express high levels of CD47[22]; Supplementary Fig. 8a. Indeed, compared to control groups, RAW264.7 macrophages expressing SIRPα Shp2-*i*SNAP exhibited a significantly increased capability in engulfing opsonized RBCs, accompanied by a transient FRET response, which returned to the basal level once the RBCs were completely engulfed (Fig. 2f, g, Supplementary Movie 3 and Supplementary Fig. 8b, c). These results indicate that the rewiring of CD47 signaling by SIRPα Shp2-*i*SNAP can promote the phagocytic ability of the engineered macrophages toward RBCs.

We then examined whether SIRPα Shp2-*i*SNAP can also promote the phagocytic power of primary macrophages against tumor cells expressing CD47. SIRPα Shp2-*i*SNAP, which was first confirmed to express correctly in primary mouse bone marrow-derived macrophages (BMDMs), is activatable by CD47-coated beads (Supplementary Fig. 9). We targeted two types of cancer cells natively expressing high levels of CD47 and tumor-specific antigens (TSAs), viz., Toledo, a human non-Hodgkin's lymphoma cell line representing hematologic cancer that expresses CD20, and DLD1, a human colon cancer cell line representing non-hematopoietic solid tumor that expresses EGFR (Supplementary Fig. 10). The engineered BMDMs led to a rapid engulfment of engaged Toledo cells opsonized by anti-CD20 antibody, accompanied by a locally activated and transient FRET

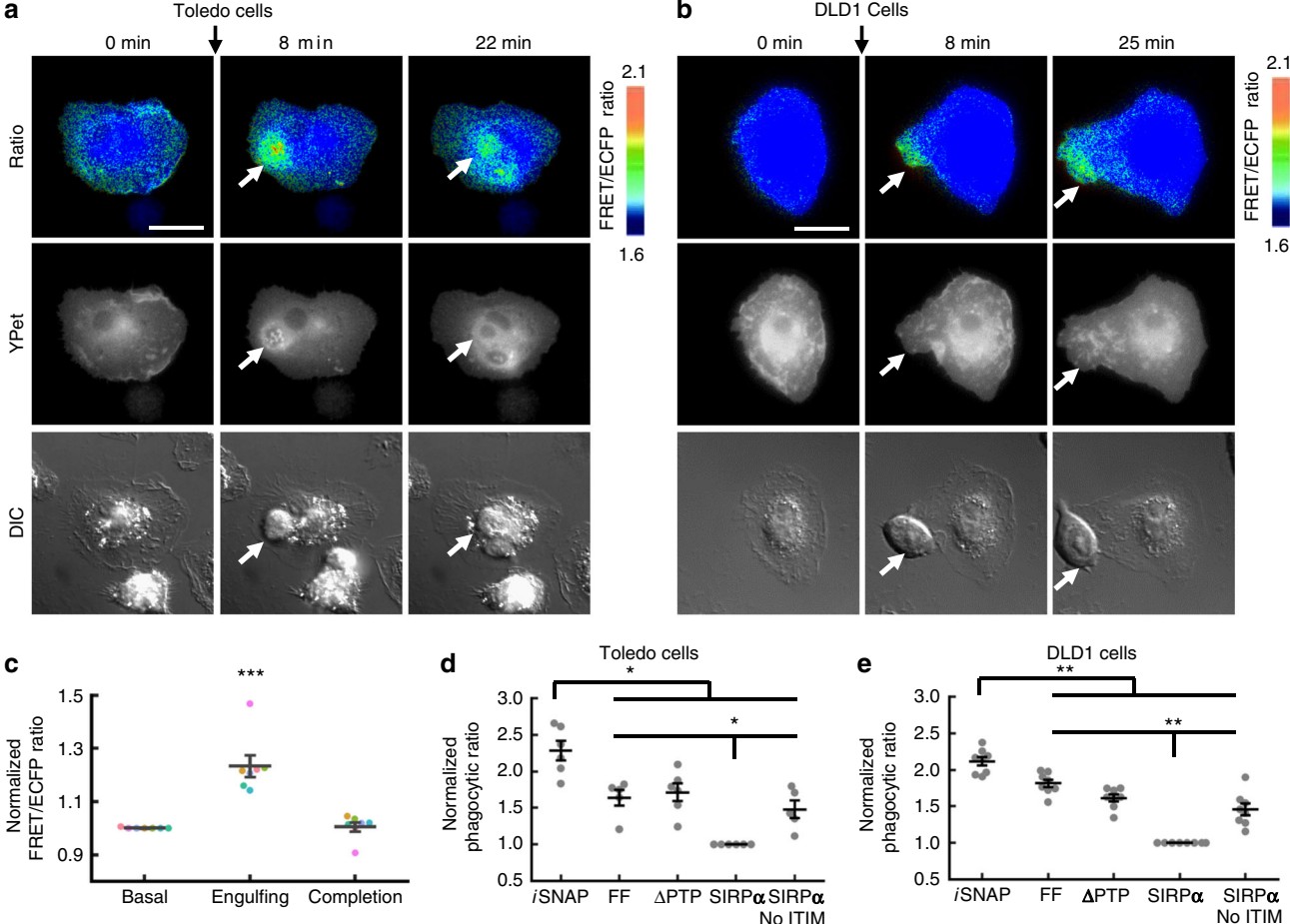

**Fig. 3** The activation and function of SIRPα Shp2-*i*SNAP on phagocytosis of cancer cells. **a** and **b** Images of a representative BMDM expressing SIRPα Shp2-*i*SNAP before and after incubation with 10 μg ml⁻¹ rituximab-opsonized Toledo cells **a** or 2 μg ml⁻¹ cetuximab-opsonized DLD1 **b** at indicated time points. **c** The FRET ratio of SIRPα Shp2-*i*SNAP in the BMDM at the region around the engaging Toledo cell before, during and after phagocytosis ($n = 7, 7, 7$). The same colored dots represent data obtained from the same cell. **d, e** Dot plot of normalized phagocytic rate (mean ± s.e.m.) of macrophages expressing different constructs as described in Fig. 2 against rituximab-opsonized Toledo **d** ($n = 6, 5, 6, 6, 5$) or cetuximab-opsonized DLD1 cells **e** ($n = 8, 8, 8, 8, 8$). *$P < 0.05$, **$P < 0.01$, ***$P < 0.001$ (two-sided Mann–Whitney $U$-test adjusted for multiple group comparison). *Scale bars*, 20 μm

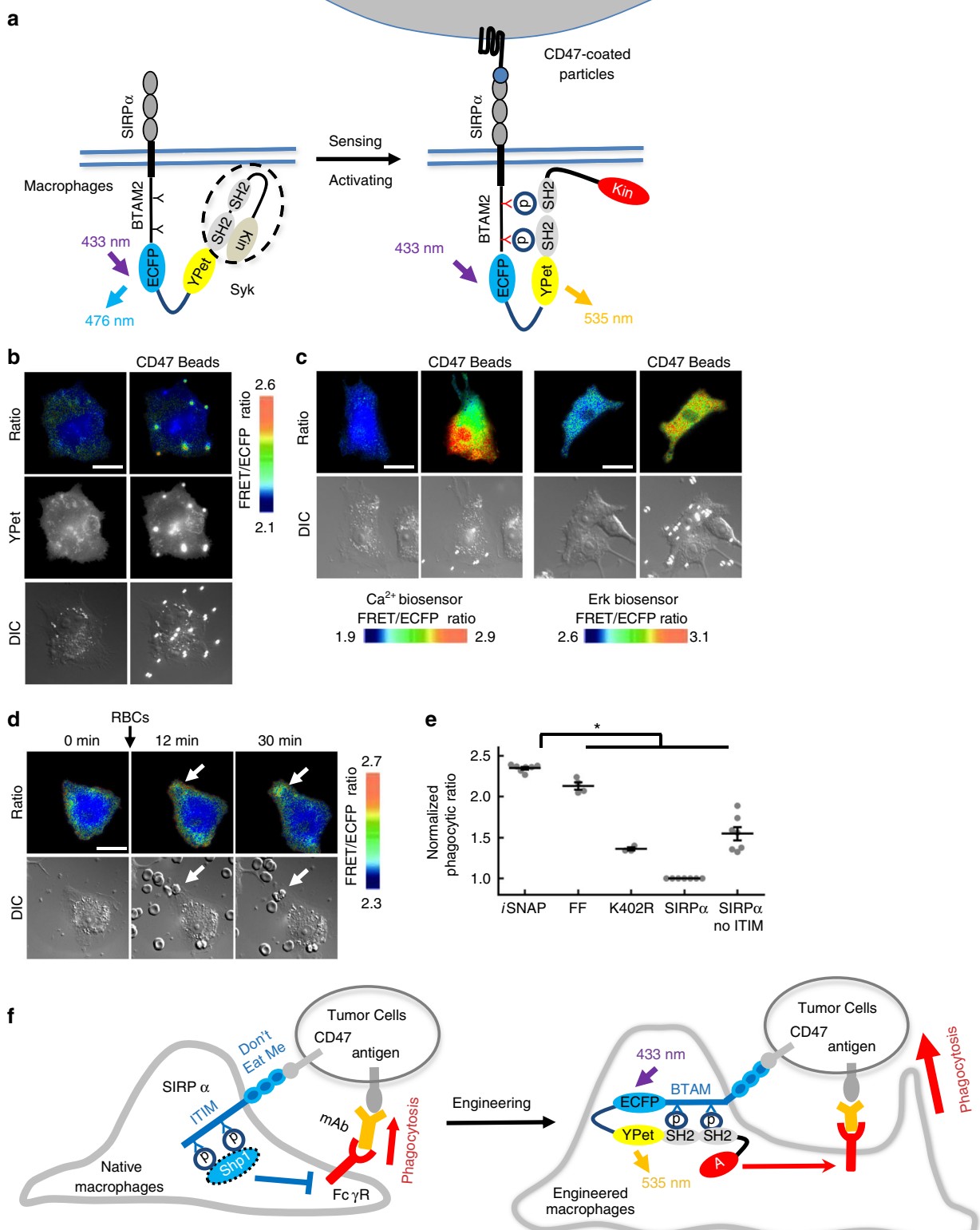

**Fig. 4** The activation and function of SIRPα Syk-*i*SNAP in macrophages. **a** Schematic representation of the drawing of the SIRPα Syk-*i*SNAP and its putative activation mechanism upon CD47 engagement. **b–c** Images of RAW264.7 macrophages expressing **b** SIRPα Syk-*i*SNAP or **c** non-fluorescent SIRPα Syk-*i*SNAP together with Ca²⁺ or Erk FRET biosensors, before and after incubation with CD47-coated beads. **d** Images of phagocytosis of opsonized RBCs by a representative RAW264.7 macrophage expressing SIRPα Syk-*i*SNAP. **e** Dot plot of normalized phagocytic rate (mean ± s.e.m.) of RAW264.7 macrophages expressing SIRPα Syk-*i*SNAP or control constructs against opsonized RBCs (K402R, a kinase-dead mutant K402R in Syk kinase domain; other constructs are the same as described in Fig. 3) ($n = 7, 4, 3, 7, 7$). **f** Schematic drawing of the engineered macrophage for mAbs-guided cancer cell eradication. In native macrophages, pro-phagocytic activity mediated by the antigen-recognizing antibody and FcγR is inhibited by CD47-SIRPα signal pathway via the recruitment of negative regulator Shp1. In engineered macrophages, the anti-phagocytic signal of CD47-SIRPα axis is rewired by SIRPα *i*SNAPs to promote phagocytic activities for the cancer cell eradication *$P < 0.05$ (two-sided Mann–Whitney $U$-test adjusted for multiple group comparison). *Scale bars*, 20 μm

signals, which returned to the basal level after the completion of Toledo phagocytosis (Fig. 3a, c and Supplementary Movie 4). In contrast, macrophages with mutant iSNAP showed sustained local FRET increase with failed Toledo phagocytosis (Supplementary Movie 4 and Supplementary Fig. 11), suggesting a correlation between FRET signal and phagocytosis outcomes. The mutation of sensing tyrosines or PTP domain in SIRPα Shp2-iSNAP caused a significant reduction of the FRET responses and of the phagocytic activity of the engineered BMDMs (Fig. 3c, d and Supplementary Figs 12–14). Similar results can be observed in engineered BMDMs when they engaged with DLD1 cells opsonized by anti-EGFR antibody (Fig. 3b, e and Supplementary Figs 15 and 16). Our results hence indicate that the rewiring of CD47 signaling in macrophages by SIRPα Shp2-iSNAP can promote the antibody-mediated tumor cell phagocytosis.

**SIRPα Syk-iSNAP rewires CD47-SIRPα axis to Syk activation.** We further examined whether our modular approach can be extended to engineer a second type of iSNAPs employing spleen tyrosine kinase (Syk) (Syk-iSNAP)[23], with the sensing peptide derived from the immunoreceptor tyrosine-based activation motif (ITAM) of FcγRIIA followed by the same FRET pair and the full-length human Syk. Syk-iSNAP was also fused to SIRPα to develop SIRPα Syk-iSNAP (Fig. 4a). CD47-coated beads induced the FRET response of the SIRPα Syk-iSNAP (Fig. 4b). Intracellular $Ca^{2+}$ and Erk activation was detected by the $Ca^{2+}$ and Erk FRET biosensors in RAW264.7 macrophages engineered with the non-fluorescent SIRPα Syk-iSNAP (Fig. 4c), but not its kinase-dead mutant (K402R) (Supplementary Fig. 17). SIRPα Syk-iSNAP in engineered RAW264.7 macrophages could also be activated by opsonized RBCs to cause enhanced phagocytic activity, mediated by the tyrosine phosphorylation and the kinase domain of the iSNAP (Fig. 4d, e and Supplementary Fig. 18). Switching the sensing tyrosine peptide to sequences derived from the ITAM motif of CD3ε and FcRγ also allowed the engineering of other Syk-iSNAPs that enhanced the macrophage phagocytosis (Supplementary Fig. 19). Therefore, with the modular design strategy, we have successfully engineered both phosphatase- and kinase-based iSNAPs capable of sensing, rewiring the 'don't eat me' CD47 signaling and reprogramming macrophages for enhanced phagocytic activities against opsonized tumor cells (Fig. 4f).

## Discussion

Our current generation of iSNAPs is based on a naturally occurring auto-inhibition mechanism within Shp2 or Syk, with the intramolecular SH2 domains in the inactive state masking the catalytic domains[23, 24]. A large number of enzymes, e.g., Zap70, Src family kinases (SFKs), FAK and Shp1, are regulated by similar auto-inhibition mechanisms, which can be utilized to engineer new iSNAPs capable of rewiring different signals and reprogramming cellular functions immediately upon receiving target signals[23, 25, 26]. This iSNAP design of fast-acting protein modules is in contrast to typical genetic circuits based on cascades of transcriptional regulation, which carry out functions with long time delays. We have also utilized this strategy to successfully engineer Shp1 and Shp1/2 hybrid iSNAPs (Supplementary Fig. 4). Synthetic protein structures have been successfully created by rational designs and modular recombination of naturally interacting domains[27, 28]. As such, our modular design allows the engineering of additional iSNAPs based on these synthetic structures. The domain components or building blocks can also be subjected to directed evolution for optimization of their functionality and orthogonality to yield high sensitivity/specificity with minimal interference of endogenous cell signaling. FRET signals of iSNAPs will provide spatiotemporal readings and serve

as a reporter for the calibration and optimization of the iSNAP functionality.

The antitumor efficacy of mAbs and macrophages can be hindered by high expression of anti-phagocytic CD47 on tumor cells. Synergistic treatment with CD47-blocking antibody and tumor-specific mAbs to promote phagocytosis and tumor eradication became an important dual-pronged strategy[10, 11, 29]. However, undesirable hemolysis of RBCs presents a major problem of anti-CD47-blocking antibodies, since CD47 is a key self-protection signal of RBCs from the macrophage phagocytosis[13, 22]. Our engineered macrophages can be specifically directed against tumor cells, but not RBCs, by TSA-recognizing antibodies to minimize off-target effects (Supplementary Fig. 20). As our iSNAPs can rewire the anti-phagocytic CD47-SIRPα signaling toward pro-phagocytic actions, CD47 on tumor cells actually promotes phagocytosis of engineered macrophages. This rewiring of CD47 signaling to Shp2 or Syk activation can also lead to the macrophage conversion toward active M1 polarization via NF-κB activation[30]. Antibody-mediated phagocytosis of cancer cells by macrophages could further initiate an antitumor T-cell immune response via antigen-presenting function of macrophages[31].

Consistent with the potent capability of SFKs in phosphorylating SIRPα[15, 32], the SFKs inhibitor PP1 markedly suppressed the FRET response of SIRPα Shp2-iSNAP (Fig. 2d and Supplementary Fig. 21). In turn, SIRPα Shp2-iSNAP provides a positive feedback loop to promote activation of SFKs, which are the main mediators of FcγR-induced phagocytosis[33]. Indeed, a FRET-based Kras-Src biosensor showed enhanced Src activity at the local regions, where beads coated by both IgG and CD47 are engaged with SIRPα Shp2-iSNAP engineered macrophages, compared to control groups using IgG-only beads or macrophages expressing SIRPα (Supplementary Fig. 22). These results suggest that SIRPα Shp2-iSNAP can rewire SIRPα signaling via Src kinase. Interestingly, Shp2-iSNAP without fusion to SIRPα failed to respond to CD47-coated beads in RAW264.7 macrophages, suggesting that the CD47-induced phosphorylation of SIRPα recruits mainly Shp1, but not Shp2, in native macrophages[22, 34] (Supplementary Fig. 23).

While the role of Syk in promoting macrophage phagocytosis is relatively clear[35–37], it remains less elucidated on how Shp2 positively contributes to phagocytosis process. Although both Shp1 and Shp2 belong to the SH2 domain-containing protein tyrosine phosphatases, Shp1 and Shp2 are generally considered as negative and positive regulators, respectively[17, 18]. In fact, in macrophages, interruption of Shp2 impairs the Syk-mediated signaling, leading to the reduction of phagocytic capabilities[20]. Shp2 may also positively regulate the phagocytic activities of macrophages via several pathways: (1) suppressing Csk to reduce the phosphorylation of inhibitory tyrosine site at C-terminal tail of SFKs for the SFK activation; (2) activating Erk and consequently ROS production[21]; (3) dephosphorylating the neighboring endogenous SIRPα and its recruitment of the negative Shp1[22, 38]; and (4) enhancing the myosin light-chain phosphorylation and contractility of actin cytoskeleton by dephosphorylating pY722 of Rho kinase (ROCK II)[39].

Many therapeutic mAbs have been developed to target different types of tumors with TSAs[6]. Our engineered macrophages with iSNAPs can be integrated with established antibody-based immunotherapy to deploy a readily switchable antibody interface for the precise targeting and efficient eradiation of various types of tumors expressing TSA and CD47. Our modular approach also allows a simple change of the iSNAP extracellular and intracellular domains to direct different input signals toward designed output functions of the macrophages. Replacing SIRPα extracellular domain with a single-chain variable fragment (ScFv) can extend our iSNAP application to T-cell engineering. Similar

approaches using *i*SNAPs could logically be applied to rewire inhibitory checkpoint signaling of adaptive immune systems for the promotion of immunotherapy and vaccine efficacy. Thus, we have engineered and characterized a new class of modular *i*SNAP proteins with dual sensing and activating functions, with their immense theranostic potential demonstrated by the proof-of-concept application in macrophage engineering for cancer immunotherapy.

## Methods

**Reagents and cell culture.** Fetal bovine serum was obtained from Atlanta Biologicals (Lawrenceville, USA). Rat recombinant PDGF BB and PP1 were from Sigma Aldrich (Milwaukee, USA). Rabbit anti-human RBC antibody (ab34858) was from Abcam (Cambridge, USA). The plasmid encoding the human SIRPα was a generous gift from Dr. Umemori at Department of Biological Chemistry, University of Michigan Medical School. FDA-approved therapeutic antibodies rituximab (anti-CD20, human IgG1) and cetuximab (anti-EGFR, human IgG1) were obtained from Dr. Dahl at VA Medical Center, San Diego. Cell line HEK293T (human embryonic kidney 293T cell line), MEF, RAW264.7 (mouse macrophage cell line), DLD1 (human colon cancer cell line), Toledo (human non-Hodgkin's B cell lymphoma cell line) and L929 cell line (murine aneuploid fibrosarcoma cell line) were from American Tissue Culture Collection (Manassas, VA), with the authentication and the verification of the absence of mycoplasma contamination. These cells were cultured in ATCC recommended conditions in a humidified incubator of 95% $O_2$ and 5% $CO_2$ at 37 °C.

**Human samples.** Normal human peripheral blood samples were obtained from VA San Diego Medical Center, San Diego, CA with an IRB-approved protocol (VA San Diego IRB# H150008). Informed consent was obtained from all human participants.

**Construction of plasmids.** The Shp2- or Shp1-*i*SNAP was constructed by fusing DNA sequences encoding BTAM peptide (GGGGDITYADLNLPKGKKPAPQAA EPNNHTEYASIQTS derived from ITIM of SIRPα)[40], ECFP (Forward primer 1: 5′-CAACCATACCGAATATGCGAGCATTCAGACCAGCGGCGGGTCTGGCG GGACAAT-3′, forward primer 2: 5′-CAAAAAACCGGCCGCCGCAGGCGGCGG AACCGAACAACCATACCGAATATGCGAGC-3′, forward primer 3: 5′-ATA TTACCTATGCGGATCTGAACCTGCCGAAAGGCAAAAAACCGGCGCGC AGG-3′, forward primer 4: 5′-CGATGGGATCCTGGCGGCGGCGGCGATATTA CCTATGCGGATCTGAACC-3′, reverse primer: 5′-ACTGCATGCGGCGGCGG TCACGAACTCC-3′), EV linker (116aa)[41] (forward primer: 5′-GTACAAGGCA TGCGAGCCTGCCAGGGGTACCAG-3′, reverse primer: 5′-GCTTGGTCGACA GGGACATCTGGTCCGGAACC-3′), YPet (forward primer: 5′-CTAAAGTCGA CATGTCTAAAGGTGAAGAATTATTCAC-3′, reverse primer: 5′-ACTGAGCTC CCCGCCTTTGTACAATTCATTCATACCCTG-3′), and the truncated human Shp2 (aa 1-532) (forward primer: 5′-ACTGAGCTCATGACATCGCGGAGATG GTTTC-3′, reverse primer: 5′-ACTGAATTCTTACTACTCTTCTTCAATCCTGC GCTG-3′) or Shp1 (aa 1-525) (forward primer: 5′-ACTGAGCTCATGGTGAGG TGGTTTCACCGAG-3′, reverse primer: 5′-ACTGAATTCTTACTAGACCTCCA GCTTCTTCTTAGTGG-3′) in pcDNA3.1 (Invitrogen) vector using BamHI/EcoRI restriction sites with 6×His tag in front of start codon of the BTAM coding sequence. The Syk-*i*SNAPs were constructed by fusing DNA sequences encoding BTAM2 peptides (GGYMTLNPRAPTDDDKNIYLTLPPN, oligonucleotides: 5′-ATTACCTGGTCCGTCAGGTCTCGGATCCAGGCGGCTACATGACTCTGA ACCCCAGGGCACCTACTGACGATGATAAAAACATCTACCTGACTCTTCC TCCCAACGGTACCGGCGGTGAGACCAGCTCACATCACCCGGGA-3′ derived from ITAM of FcγR IIA, or GVYTGLSTRNQETYETLKHE, oligonucleotides: 5′-GAAGATTACCTGGTCCACGTCAGGTCTCGGATCCAGGTGTTTACACGG GCCTGAGCACCAGGAACCAGGAGACTTACGAGACTCTGAAGCATGAGG GTACCGGCGGTGAGACCAGCTCACATCACCCGGGA-3′ derived from ITAM of FcRγ, or MPDYEPIRKGQRDLYSGLNQR oligonucleotides: 5′-CGTCAGGTC TCGGATCCACCAGACTATGAGCCCCATCCGGAAAGGCCAGCGGGACCTGT ATTCTGGCCTGAATCAGAGAGGCGGGTCTGGCGGGACAGGTACCGGCG GTGAGACCAGCTCACATCACCCGGGA-3′ derived from ITAM of CD3ε)[40], PCR product of ECFP (forward primer: 5′-AGATCGGTCTCGGCGGTATGGT GAGCAAGGGCGAGG-3′, reverse primer: 5′-GAGTTCGTGACCGCCGCCAT GCgGAGACCAGCTC-3′), EV linker (oligonucleotides: 5′-CCTGGTCCGTCAGG TCTCCATGCGAGCCTGCCAGGAGCGCAGGCGGATCAGCTGGAGGGTCT GCAGGGGGTAGTGCAGGTGGCTCAGCTGGCGGGAGCGGCTCAGCTGGG GGATCTGCTGGTGGCAGTACCTCAGCAGGCGGTAGCGCCGGAGGTTCTG CTGGTGGCTCCGCAGGAGGGTCTGCAGGCGGTTCCGGGAGTGCAGGTG GATCTGCAGGTGGGTCAACAAGTGCTGGTGGATCCGCAGGAGGTTCAGC AGGCGGGAGTGCTGGAGGCTCTGCAGGCGGTAGCGGGAGTGCCGGTGG CAGCCAGGGGGAAGCACTAGTGCTGGAGGCAGTGCAGGTGGCAGCGC AGGAGGCTCTGCCGGGGGGAAGCGCCGGGGGCTCCGGACCAGATGTCCC TGTCGACGAGACCAGCTCACATCA-3′), PCR product of YPet (forward primer: 5′-GATCGGTCTCGTCGACTCTAAAGGTGAAGAATTATTCACTG-3′, reverse primer: 5′-AGGGTATGAATGAATTGTACAAAGAGACCAGCTC-3′,

and PCR product of full-length human SYK (forward primer: 5′-GATCGGTCT CTACAAAGGCGGGGAgctcGCCAGCAGCGGCATGGCTGACAGCGCCAAC CAC-3′, reverse primer: 5′-CAATTACTACTATGACGTGGTGAACTAAGAA TTCGAGACGAGCTC-3′) into BamHI/ EcoRI restriction sites of pcDNA3.1 vector using Golden Gate assembly, respectively. FF mutant of Syk-*i*SNAP was constructed by replacing oligonucleotides coding wild-type peptide with oligonucleotides: 5′-ATTACCTGGTCCGTCAGGTCTCGGATCCAGGCGGCTTTATGA CTCTGAACCCCAGGGCACCTACTGACGATGATAAAAACATCTTTCTGAC TCTTCCTCCCAACGGTACCGGCGGTGAGACCAGCTCACATCACCCGGG-3′ carrying Y to F mutations. For the SIRPα fused *i*SNAPs, the PCR product of gene sequence encoding human SIRPα (aa 1-425) (forward primer: 5′-AGCCCAAGCT TGCCACCATGGAGCCCGCC-3′, reverse primer: 5′-TCGGGGATCCGAATTT GTGTCCTGTGTTATTTC-3′) was inserted in 5′-of *i*SNAP coding sequences using HindIII/BamHI restriction sites. Deletion and point mutations of *i*SNAP were generated using QuikChange Site-Directed Mutagenesis Kit (Agilent Technologies) with primer sets: 5′-CAGGAGTGCAAACTTCTCTACAGCCAGCGCA GGATTGAAGAAGAG-3′/5′-CTCTTCTTCAATCCTGCGCTGGCTGTAGAGA AGTTTGCACTCCTG-3′ for SIRPα Shp2-*i*SNAP ΔPTP; 5′-GTGAAAACCGTG GCTGTGAGGATACTGAAAAACGAGGCCA-3′/5′-TGGCCTCGTTTTTCAGT ATCCTCACAGCCACGGTTTTCAC-3′ for SIRPα Syk-*i*SNAP K402R. For SIRPα-YPet and SIRPα-no ITIM, PCR products of human SIRPα coding sequence (forward primer: 5′-AGCCCAAGCTTGCCACCATGGAGCCCGCC-3′, reverse primer: 5′-CGTGGAATTCTGTTCCGCCAGATCCGCCCTTCCTCGGGACCTG-3′ for aa 1-504, full-length); or reverse primer: 5′-CGTGGAATTCTGTTCCGCCA GATCCGCCATTTGTGTCCTGTGTTATTTC-3′ for aa 1-425, no ITIM) and PCR product of YPet (forward primer: 5′-TCCGGAATTCATGTCTAAAGGTGAAG AATTATTCACTG-3′, reverse primer: 5′-ACAGACCTCGAGTCATTTGTACA ATTCATTCATACCCT-3′) were inserted into HindIII/ Xho I restriction sites of pcDNA3.1. ECFPT65A/W66A and YPet Y66A mutations were done with primer sets 5′-CGTGACCACCCTGGCCGCGGGCGTGCAGTGC-3′/ 5′-GCACTGCAC GCCCGCGGCCAGGGTGGTCACG-3′, 5′-TGGCCAACCTTAGTCACTACTTT AGGTGCTGGTGTTCAATGTTTTG-3′/5′-CAAAACATTGAACACCAGCACCT AAAGTAGTGACTAAGGTTGGCCA-3′, respectively. The constructed plasmids were confirmed by restriction enzyme digestion and DNA sequencing.

**iSNAP expression and purification.** HEK cells transfected with wild-type *i*SNAPs or their mutants were washed with cold PBS and then lysed in buffer containing 50 mM Tris-HCl pH 7.5, 100 mM NaCl, 1 mM EDTA, 0.2 mM PMSF, 0.2% Triton X-100 and a protease inhibitor cocktail tablet (Roche). Lysates were centrifuged at 10,000 $g$ at 4 °C for 10 min. Supernatants were incubated with Ni-NTA agarose (Qiagen) to capture the desired protein products via their 6-His tag at the N-termini, which were then washed with 50 mM Tris-HCl pH 7.5, 100 mM NaCl, 10 mM imidazole. *i*SNAP proteins were eluted in a buffer containing 50 mM Tris-HCl pH 7.5, 100 mM NaCl, and 100 mM imidazole.

**In vitro kinase assays.** Purified proteins were dialyzed overnight at 4 °C in kinase buffer (50 mM Tris-HCl pH 8.0, 100 mM NaCl, 10 mM $MgCl_2$, and 2 mM dithiothreitol). Fluorescence emission spectra of the purified *i*SNAP proteins (50 nM) were measured in 96-well plates with excitation wavelength of 430 nm using a fluorescence plate reader (TECAN, Sapphire II). To detect FRET changes of the *i*SNAPs, emission ratios of acceptor/donor (526 nm/478 nm) were measured at 30 °C before and after the addition of 1 mM ATP and 100 nM active recombinant Src (Millipore).

**In vitro phosphatase activity assays.** After kinase assay, phosphatase activity was measured by adding fluorogenic 6,8-difluoro-4-methylumbelliferyl phosphate (DiFMUP; Invitrogen) as the substrate. In brief, each reaction contained 50 mM Tris pH 8, 100 mM NaCl, 10 mM $MgCl_2$, 2 mM dithiothreitol, 100 nM Src kinase, 50 nM *i*SNAP proteins and 50 μM DiFMUP in a total reaction volume of 100 μl in a well of 96-well plates. Reactions were initiated by the addition of DiFMUP, followed by measuring the fluorescence signal of the reaction product, 6,8-difluoro-4-methylumbelliferone, at an excitation wavelength of 355 nm and an emission of 460 nm with a plate reader (TECAN, Sapphire II). The intensity slope of the fluorescent product over time was calculated and compared. Our statistics were performed with two-tailed Student T-test.

**Immunoblotting.** After in vitro assays, the *i*SNAP proteins were resolved by SDS-PAGE. The proteins were then transferred onto a nitrocellulose membrane and blocked with 5% bovine serum albumin in TTBS buffer (50 mM Tris-HCl, 145 mM NaCl, 0.05% Tween-20, pH 7.4) for 2 h at room temperature (RT). Membranes were further incubated with primary antibodies overnight at 4 °C, washed and then incubated with horseradish peroxidase (HRP) -conjugated secondary antibodies for 2 h at RT. Signals were detected using SuperSignal Western Pico or Femto ECL Kit (Pierce). Monoclonal anti-phosphotyrosine pY20 antibody was from BD Transduction laboratory (61000, 1:1000 dilution). And polyclonal anti-GFP antibodies (sc-8334, 1:1000 dilution) as well as HRP-conjugated secondary antibodies (sc-2004, sc-2005, 1:2000 dilution) were purchased from Santa Cruz Biotechnology.

**Microscopy image acquisition and analysis**. Glass-bottom dishes (Cell E&G Inc.) were coated with 10 µg ml$^{-1}$ fibronectin (Sigma) overnight at 37 °C. Transfected or electroporated cells were plated onto these dishes overnight in medium containing 0.5% fetal bovine serum (FBS) before imaging. During imaging, cells were maintained in medium containing 0.5% FBS with 5% CO$_2$ supplement at 37 °C and a few frames of images were acquired to obtain the basal level before adding stimulation. For PDGF stimulation, PDGF was added into medium to reach final concentration 10 ng ml$^{-1}$. For RBCs, Toledo and DLD1 cells stimulation, those cells were washed with PBS plus 0.4% bovine serum albumin (BSA), and opsonized with rabbit anti-human RBC IgG (Abcam, ab34858, 5 µg ml$^{-1}$), rituximab (anti-CD20 mAb, 10 µg ml$^{-1}$), cetuximab (anti-EGFR mAb, 2 µg ml$^{-1}$) or trastuzumab (anti-HER2 mAb, 2 µg ml$^{-1}$) for 60 min at RT, respectively. After washing with PBS plus 0.4% BSA, cells were re-suspended in medium with 0.5% FBS and applied into dishes. Images were obtained by a Nikon eclipse Ti inverted microscope equipped with a cooled charge-coupled device camera (Cascade 512B; Photometrics) with 512 × 512 resolution using MetaFluor 6.2 software (Universal Imaging). The following filter sets (Chroma) were used in our experiments for FRET imaging: a dichroic mirror (450 nm), an excitation filter 420/20 nm, an ECFP emission filter 475/40 nm and a FRET emission filter 535/25 nm. The excitation filter for ECFP at 420 ± 20 nm was specifically selected to shift toward lower wavelength away from the peak excitation spectrum of ECFP to reduce the cross-excitation of the FRET acceptor YPet, which has significantly higher brightness than ECFP. This filter selection can minimize the effect of bleed-through on the FRET channel. The fluorescence intensity of non-transfected cells was quantified as the background signal and subtracted from the ECFP and FRET signals of transfected cells. The pixel-by-pixel ratio images of FRET/ECFP were calculated based on the background-subtracted fluorescence intensity images of ECFP and FRET. These ratio images were displayed in the intensity modified display mode in which the color and brightness of each pixel is determined by the FRET/ECFP ratio and ECFP intensity, respectively. The emission ratios were quantified by MetaFluor software and our statistics were performed with two-tailed Student T-test using Excel (Microsoft).

**Recombinant human CD47 production and coating on beads**. Plasmid encoding the extracellular domain of human CD47 was a gift from Dr. Dennis E. Discher (Molecular and Cell Biophysics Laboratory, University of Pennsylvania). The plasmid was transfected into HEK293T cell using Lipofectamine 2000 (Invitrogen). After 36 h culture, medium containing secreted CD47-CD4d3 was concentrated using a 10K MWCO Amicon (Millipore), and CD47-CD4d3 was biotinylated at the C terminus using a biotin-protein ligase (Avidity, LLC) and dialyzed against PBS for 24 h. The biotinylated CD47-CD4d3 was affinity purified using monomeric avidin (Promega) and dialyzed against PBS for 24 h. Streptavidin-coated polystyrene beads of 4 µm diameter (Spherotech) were washed in PBS plus 0.4% BSA, and then incubated with biotinylated CD47 at RT for 30 min, followed by 3× wash and re-suspension in cell culture medium with 0.5% FBS. For IgG coating, streptavidin-coated polystyrene beads were incubated with rabbit anti-streptavidin IgG (Abcam, ab6676, 1:3000 dilution) at RT for 30 min.

**Flow cytometry**. For soluble CD47-binding assay, HEK293T cells were transfected with SIRPα Shp2-*i*SNAP, its mutants, SIRPα-YPet or truncated SIRPα (no ITIM motif) fused to YPet by Lipofectamine 2000 (Life technologies). After 36 h culture, transfected HEK293T cells were detached by treatment of PBS with 10 mM EDTA and non-specific surface residues were blocked for 10 min by PBS plus 0.4% BSA. Cells were then incubated with biotinylated CD47 at room temperature (RT) for 30 min, washed and then incubated with PE-conjugated streptavidin (Life technologies, SA10041, 1:1000 dilution) at RT for 30 min.

For the measurement of surface CD47, CD20 and EGFR expressions, RBCs, Toledo and DLD1 (detached by treatment of Accutase, Innovative Cell Technologies, Inc.) cells were washed with PBS plus 0.4% BSA, and incubated with mouse anti-human CD47 IgG (Abcam, ab3283, 1:1000 dilution), rituximab (1:1000 dilution) or cetuximab (1:1000 dilution) at RT for 60 min, respectively. After wash with PBS plus 0.4% BSA, cells were incubated with secondary antibodies: Alexa Fluor 488-conjugated goat anti-mouse antibody for CD47, (Life technologies, A32723, 1:1000 dilution), PE-conjugated rabbit anti-human antibody (Abcam, ab98596, 1:500 dilution) for CD20 and EGFR, at RT for 60 min.

For RBC opsonization, RBCs were washed with PBS plus 0.4% BSA, incubated with rabbit anti-hRBC IgG (5 µg ml$^{-1}$) for 60 min at RT, followed by washing and incubation with Alexa Fluor 647-conjugated goat anti-rabbit secondary antibody (Life technologies, A32733, 1:1000 dilution) at RT for 60 min.

Cells were analyzed using flow cytometry after antibody staining. The acquisition configurations using a BD Accuri$^{TM}$ C6 (BD Immunocytometry Systems) for different FPs and dyes are listed as following: YPet (Ex:488, Em:530/30 nm); PE (Ex:488, Em:575/26 nm); Alexa Fluor 647 (Ex:640, Em:660/20 nm).

**Differentiation of bone marrow-derived macrophages**. Bone marrow from wild-type C57BL/6 12–20 weeks female mice were harvested from freshly isolated femurs, tibiae and humeri. After removal of connective tissues and muscles, bone marrow cells were flushed out and single-cell suspensions were made by pipetting and passing bone marrow through a sterile 70 µm filter (BD Falcon). Remaining

RBCs were lysed by ACK buffer (0.15 M NH$_4$Cl, 10 mM KHCO$_3$, 0.1 mM EDTA). Macrophages were differentiated by incubating bone marrow cells for 7 days with complete RPMI 1640, supplemented with 10% L929-conditioned medium (containing M-CSF). Macrophages were harvested after 15-min incubation with 10 mM EDTA. All animal work was approved by the USCD Animal Care Program (S000227M, UCSD Animal Care Program/IACUC).

**Electroporation of macrophages and phagocytosis assay**. *i*SNAPs were introduced into RAW 264.7 macrophages and BMDMs by electroporation. Briefly, 4 × 10$^6$ RAW 264.7 or BMDMs were re-suspended in 240 µl of RPMI 1640 with 20 µg plasmids, and then electroporated (250 V, 950 mF, ∞ Ω). For phagocytosis assays, transfected macrophages were plated in 35 mm dishes at 20% confluency. RBCs, Toledo and DLD1 (detached by treatment of Accutase) cells were washed 3 times with PBS, and incubated with CellTracker™ Deep Red dye (Life technologies) in PBS (1 µM for RBCs labeling, 0.3 µM for Toledo and DLD1 cells labeling) for 30 min at RT followed by three times of wash with PBS. RBCs were added to RAW264.7 macrophages at a ratio of 10:1 and allowed to incubate at 37 °C for 30 min in the presence of rabbit anti-hRBC IgG (5 µg ml$^{-1}$). Free RBCs were removed by washing with PBS plus 0.4% BSA; bounded RBCs were lysed by adding ACK buffer (NH$_4$Cl 31 mM, KHCO$_3$ 2 mM, EDTA 20 µM) for 2 min. Toledo or DLD1 cells were added to BMDMs at a ratio of 10:1 and allowed to incubate at 37 °C for 4 h in the presence of rituximab (10 µg ml$^{-1}$) for Toledo cells or cetuximab (2 µg ml$^{-1}$) for DLD1 cells, respectively. Free Toledo or DLD1 cells were removed by washing with PBS; bounded cells were dissociated by incubation with EDTA 10 mM at 37 °C for 20 min and dissociation efficiency was confirmed under microscope. Macrophages were detached by scratching with plastic lifter and kept in suspension. Engineered macrophage population defined as YPet$^+$ cells were measured by flow cytometry and gated, and the signal of ingested cells (Deep Red$^+$, Ex:640 nm, Em:660/20 nm) was determined within engineered macrophage population. Expression level of SIRPα Shp2-*i*SNAP or control constructs represented by the intensity of YPet signal (Ex: 488 nm, Em: 575/26 nm) were recorded simultaneously. Doublets discrimination flow cytometry was used to further distinguish internalized from externally bound tumor cells[42, 43]. The phagocytic potential (mean ± s.e.m. from multiple independent measurements) of different macrophage groups were normalized to the level of phagocytosis of target cells by macrophages expressing SIRPα-YPet. Each dot represents mean normalized phagocytic rate from an individual experiment, error bars represent s.e.m.

**Statistical analysis**. Statistical analysis was performed by two-sided Mann–Whitney *U*-test in MATLAB, with the *P*-values adjusted for comparison of multiple groups. Statistics of Fig. 1d was done by exact randomized permutation test. All the center values are mean ± s.e.m. A significant difference was determined by *P* value < 0.05. All the experiments were replicated at least three times and represented biological replicates.

**Data availability**. The DNA sequences of *i*SNAP have been deposited into NCBI GenBank (Shp2-*i*SNAP: MF434748, SIRPa Shp2-*i*SNAP: MF434749, SIRPa *i*SNAP ΔPTP: MF434750, SIRPa Syk-iSNAP: MF434751, SIRPa YPet: MF434752, SIRPa no ITIM YPet: MF434753). The data that support the findings of this study are available from the corresponding author upon request.

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

## Acknowledgements

We thank Profs. Roger Y. Tsien, Dennis Discher, Christopher K. Glass, Yury Miller and Jack Bui at UCSD for helpful discussions and invaluable suggestions, as well as plasmids and reagents. We appreciate Dr. Brian J. Dahl at Veterans Affairs San Diego Healthcare System for providing rituximab and cetuximab. The human SIRPα construct is a generous gift from Dr. Hisashi Umemori at Department of Neurology, F.M. Kirby Neurobiology Center, Boston Children's Hospital, Harvard Medical School. This work is supported by grants from NIH HL121365 and CA209629 (Y. Wang), NSF CBET1360341, DMS1361421 (Y. Wang and S.L.), UC San Diego, and Beckman Laser Institute, Inc. (Y. Wang). This research was also supported by NSF China NSFC 11428207 (Y. Wang). The funding agencies had no role in study design, data collection and analysis, decision to publish, or preparation of the manuscript.

## Author contributions

J.S., L.L., S.C. and Y. Wang designed research. J.S., L.L., C.-M.T., Y.W., M.O., J.Seong, T.K., P.W. and M.H. performed research. J.S., L.L., Y.S. and S.L. analyzed the data. X.X. and V.N. provided reagents. J.S., L.L., S.L., X.X., V.N., S.C. and Y. Wang wrote the paper.

## Additional information

**Competing interests:** The authors declare no competing financial interests.

