## [Peer Review File · Nature Communications]

Reviewers' Comments:

Reviewer #1:

Remarks to the Author:

This revised version of the paper, now submitted to Nature Communications, addresses most of my concerns from my previous review. The introduction now provides more rationale for why Shp2 activation should dampen the "don't eat me" signal and the paper more clearly points out the correlation between FRET signaling and functional outcomes. The authors also do a better job of reporting how many times various experiments were performed and provide more thorough error analysis. The ability to both sense changes in cellular signaling and initiate a response in response to these changes has the potential to be a powerful combination; the authors suggest that one could use a system such as theirs as a "digital multi-meter" for regulating signaling in a precise manner. The paper would have higher impact if this type of feedback experiment was performed, however, the results are still exciting and warrant publication in Nature Communications.

Reviewer #2:

Remarks to the Author:

I am satisfied with the response to my comments. I think the revision significantly improved the manuscript. I was most concerned with the fold changes in phagocytic activity and the authors addressed this issue adequately. I am also pleased to see a better statistical analysis of their data.

Reviewer #3:

Remarks to the Author:

The authors present a combination of a biosensor and an actuator. From a biosensor engineering perspective, there is little novelty. The authors refer to the system as a digital multimeter, which is a misnomer; it is not digital and it is not a multimeter (it only measures a single parameter).

I have seen a previous version of the manuscript and I am glad to see that the data presentation is improved. There are a couple of suggestions I can make for further improvement of data presentation:

- List the number of cells/samples in the graph per condition.
- The data in graph 2d and 3c present FRET ratios from different conditions but acquired from the same cell. Therefore, these data are connected/paired. Please use lines to connect the data points.

The manuscript needs serious rewriting. It is very concise (as an example see last paragraph of page 4 of main text, starting with "Next, we examined RAW264.7 macrophage cell line engineered with SIRP Shp2-iSNAP"), providing little background for non-experts. The rationale for each of the steps is hardly explained. Please elaborate why the experiments were performed. The manuscript does not have a clear separation of introduction, results and discussion.

Comments on the data:

-In figure 1b, a large peak (sensitized emission and possibly direct excitation of the acceptor) are visible. Hardly any CFP fluorescence is visible, which is worrying. The authors should provide convincing data that this is a change in FRET and that CFP is indeed present. One way to achieve this is to add trypsin, to abolish FRET, which should reveal a clear CFP-like component.

-What does the emission spectrum of the FF variant look like?

-To make a compelling case that the change in FRET/CFP ratio is due to FRET (in figure 1f and 2d), the authors should provide intensity traces of the CFP and YFP (FRET) channel. A bona fide FRET increase is characterized by a YFP increase and a CFP decrease

-The data presented in figure 1h is presented as ECFP/FRET, which is confusing and should be inverted for consistency. The effect of iSNAP is statistically significant, but the effect is very small and therefore not convincing.

-The responses depicted in figure 2d is convincing, and will be even more so if it is shown that these data are from the same cell.

We greatly appreciate editorial help and the reviewers' time and effort in providing this invaluable feedback to our work. We are pleased that reviewers 1 and 2 are satisfied by our revision.

In response to reviewer 3's thoughtful comments and suggestions, we have performed further image analysis, replaced 2 figures, and provided 4 more figures for reviewers. We believe we have made thorough revisions to address the reviewer's concerns. Here, we submit a list of point-to-point responses to the formatting requirements and reviewer's comments. The major corresponding modifications are highlighted in yellow color in the revised manuscript.

The point-to-point responses to reviewer 3's critiques are listed as following:

1-The authors present a combination of a biosensor and an actuator. From a biosensor engineering perspective, there is little novelty. The authors refer to the system as a digital multimeter, which is a misnomer; it is not digital and it is not a multimeter (it only measures a single parameter).

Response: The main focus of our paper is on the engineering of a new class of molecules for live cells, integrating the functional modules of FRET sensing and enzyme activating into one molecule (*integrated sensing and activating protein, iSNAP*), which is different from biosensor engineering. We agree that the "digital multimeter" analogy does not precisely reflect our system and have hence deleted the term "digital multimeter" from the manuscript.

2-List the number of cells/samples in the graph per condition.

-The data in graph 2d and 3c present FRET ratios from different conditions but acquired from the same cell. Therefore, these data are connected/paired. Please use lines to connect the data points.

Response: We are listing the number of cells/samples in the figure legends in accordance with publications in Nature Communications. We have tried to also list the number of cells/samples in each graph, but this leads to crowded appearance of the figures.

We have followed the reviewer's suggestion to connect the data acquired from the same cell, with each cell coded with a different color (new Fig. 2d and 3c). We have also tried to connect the data points from the same cell with lines (Figure 1 for reviewer). The figures appeared busy due to the large number of points and lines, so we used the color-coded dots instead, which hopefully serves the same purpose.

3.The manuscript needs serious rewriting. It is very concise (as an example see last paragraph of page 4 of main text, starting with "Next, we examined RAW264.7 macrophage cell line engineered with SIRP Shp2-iSNAP"), providing little background for non-experts. The rationale for each of the steps is hardly explained. Please elaborate why the experiments were performed. The manuscript does not have a clear separation of introduction, results and discussion.

Response: We have rewritten the manuscript, providing background and rationales of

different steps (highlighted in yellow). We have separated introduction, results, and discussion.

Comments on the data:

4-In figure 1b, a large peak (sensitized emission and possibly direct excitation of the acceptor) are visible. Hardly any CFP fluorescence is visible, which is worrying. The authors should provide convincing data that this is a change in FRET and that CFP is indeed present. One way to achieve this is to add trypsin, to abolish FRET, which should reveal a clear CFP-like component.

Response: The excitation wavelength for the *i*SNAP is 430nm, which should have minimal direct excitation of the acceptor YPet. The donor ECFP brightness is much lower than that of the acceptor YPet. Together with the energy transfer from ECFP to YPet under conditions of both before and after Src kinase addition, the ECFP emission peak is less obvious. But the peak of ECFP can be still observed if this specific zone is enlarged, particularly when the *i*SNAP is not phosphorylated and there is less FRET (Figure 2 for reviewer). Furthermore, there is a clear increase in acceptor YPet emission and a concurrent decrease in donor ECFP emission of *i*SNAP upon phosphorylation by kinase. This suggests that there is a clear and convincing FRET change of *i*SNAP upon activation, which is also consistent with the reviewer's note "A bona fide FRET increase is characterized by a YFP increase and a CFP decrease" in critique#6 as shown below.

5-What does the emission spectrum of the FF variant look like?

Response: The emission spectrum of the FF variant is similar to the wild-type *i*SNAP before it is phosphorylated (Figure 3 for reviewer)

6-To make a compelling case that the change in FRET/CFP ratio is due to FRET (in figure 1f and 2d), the authors should provide intensity traces of the CFP and YFP (FRET) channel. A bona fide FRET increase is characterized by a YFP increase and a CFP decrease

Response: We have now included intensity traces of the CFP and YFP (FRET) channels from one representative cell to indicate that there is a YFP increase and a CFP decrease for Figures 1f and 2d (Figure 4 for reviewer).

7-The data presented in figure 1h is presented as ECFP/FRET, which is confusing and should be inverted for consistency. The effect of *i*SNAP is statistically significant, but the effect is very small and therefore not convincing.

-The responses depicted in figure 2d is convincing, and will be even more so if it is shown that these data are from the same cell.

Response: Figure 1h is quantification for the FAK (focal adhesion kinase) biosensor rather than *i*SNAP. Due to the structure of binding domain (e.g. SH2 domain) for phosphorylated tyrosine substrate peptide, the design of FRET biosensors for tyrosine kinases, such as FAK and Src, results in a FRET decrease upon activation (or phosphorylation). To be consistent with the publications in FRET field, the ratio of ECFP/FRET for FAK biosensor was utilized to represent FAK activity so that an increase of kinase activity is represented by an increase in ratio¹⁻⁴. Hence we used the same ECFP/FRET ratio to quantify FAK biosensor here to be consistent with previous

FAK publications. The less dramatic effect of *i*SNAP may be due to the limited sensitivity (dynamic range) of the reporter (FAK biosensor)². Furthermore, FAK may be regulated by multiple upstream molecules with possibly a fraction of them controllable by *i*SNAP; this may have limited the net effect of *i*SNAP on FAK regulation. Nevertheless, multiple experimental repeats clearly indicate a statistically significant effect of *i*SNAP on FAK activation.

We have followed the reviewer's suggestion to connect the data acquired from the same cell in Fig. 2d, as shown in our response #2. A time course from the same cell is also shown in supplementary fig 21c.

1. Seong, J., Lu, S. & Wang, Y. Live Cell Imaging of Src/FAK Signaling by FRET. *Cell Mol Bioeng* **2**, 138-147 (2011).
2. Seong, J. et al. Detection of focal adhesion kinase activation at membrane microdomains by fluorescence resonance energy transfer. *Nature communications* **2**, 406 (2011).
3. Seong, J. et al. Distinct biophysical mechanisms of focal adhesion kinase mechanoactivation by different extracellular matrix proteins. *Proc Natl Acad Sci U S A* **110**, 19372-19377 (2013).
4. Wu, Y. et al. In-situ coupling between kinase activities and protein dynamics within single focal adhesions. *Scientific reports* **6**, 29377 (2016).

Figure 2d with color-coded dots

Figure 2d with line connections

Figure 3c with color-coded dots

Figure 3c with line connections

Figure 1b

Figure 1b black curve in the box enlarged

Figure 1b for WT *i*SNAP

b

FF *i*SNAP with the same y-axis scale as WT *i*SNAP for comparison

The black and red curves are overlapping

Only the black curve is shown.

Figure 4 for reviewer

One cell from Figure 1f, there is a clear increase of FRET(YFP channel) and a decrease of ECFP, indicating FRET efficiency increase after PDGF addition.

One cell from Figure 2d, CD47 beads induce SIRP-*i*SNAP aggregation (as shown in supplementary figure 21a), resulting an increase of both FRET and ECFP intensities, possibly due to a drastic change of structure and morphology at the proximal region of the beads which leads to the copy number change of the SIRP-*i*SNAP at the local site. The increase of FRET intensity, however, is obviously greater than that of ECFP, so the calculated FRET/ECFP ratio increases compared to before CD47 beads addition.

Reviewers' Comments:

Reviewer #3:

Remarks to the Author:

The authors have incorporated the suggested changes and the manuscript is improved.

The authors provided some figures that were only included in the rebuttal and I would like to comment on those.

In 'figure 2 for reviewer', an ECFP peak is supposed to be present. I'm not convinced.

In 'figure 4 for reviewer', the CFP/YFP traces are shown, supposedly showing CFP decrease and concomitant YFP increase. Again, I don't think these data are convincing.

Still, I think the authors did what they can do (and we have to accept that experiments are never perfect) and I feel that the manuscript should be accepted for publication under the condition that the figures discussed above (figure 2&4 for reviewer) are also presented in the main text (NOT in supplemental data) of the manuscript. In this way, the audience can judge the data for themselves.

We greatly appreciate your editorial help and the reviewer's time and effort in providing this invaluable feedback to our work. The point-to-point responses to reviewer 3's critiques are listed as following:

1-The authors have incorporated the suggested changes and the manuscript is improved. The authors provided some figures that were only included in the rebuttal and I would like to comment on those. In 'figure 2 for reviewer', an ECFP peak is supposed to be present. I'm not convinced. In 'figure 4 for reviewer', the CFP/YFP traces are shown, supposedly showing CFP decrease and concomitant YFP increase. Again, I don't think these data are convincing.

Still, I think the authors did what they can do (and we have to accept that experiments are never perfect) and I feel that the manuscript should be accepted for publication under the condition that the figures discussed above (figure 2&4 for reviewer) are also presented in the main text (NOT in supplemental data) of the manuscript. In this way, the audience can judge the data for themselves.

Response: We thank reviewer for the comments and suggestions. We followed the reviewer's suggestions and have included "figure 2&4 for reviewer" in the main figure (see new main figure 1b, 1g, 2e) with modified figure legends and main text (highlighted in yellow).